# Research on the impact of career advancement on the mental health of young workers

Hanwei Li[1][☯][¤a], Xiaoheng Wu[iD][2][☯][*][¤b]

1 School of Management, Shanghai University of Engineering Science, Shanghai, China, 2 School of Public Administration and Policy, Shanghai University of Finance and Economics, Shanghai, China

☯ These authors contributed equally to this work.
¤a Current Address: School of Management, Shanghai University of Engineering Science.
¤b Current Address: School of Public Administration and Policy, Shanghai University of Finance and Economics.
* wuxh2305@163.com

## Abstract

Mental health is an important component of overall health. Currently, the mental health issues among Chinese workers are becoming increasingly severe, especially under the backdrop of increased uncertainty in career advancement, where depression and anxiety issues among young workers are prominent. Based on the panel data from the China Family Panel Studies (CFPS) in 2018 and 2020, this paper constructs an OLS regression model to empirically analyze the impact of career advancement constraints on workers' mental health and its mechanism of action. The study finds that limited career advancement significantly increases workers' depression index. Pathways such as extended working hours, deteriorated health conditions, and distorted labor value concepts play a partial mediating role, thereby reducing mental health levels. Heterogeneity analysis shows that compared to female, low-income, and low-educated individuals, male, those with medium to high income, and higher education levels are more likely to experience a decline in mental health due to career advancement constraints. This study provides empirical evidence for understanding the impact of career advancement uncertainty on mental health and offers important insights for improving occupational mental health policies and enhancing worker welfare.

## Introduction

Mental health is a state of well-being in which individuals exhibit rational cognition, emotional stability, appropriate behavior, harmonious interpersonal relationships, and adaptability to change during their growth and development. It constitutes a vital component of overall health. The Chinese government has always attached great importance to public health and social welfare. In 2019, it released the "Healthy China Initiative (2019–2030)", urging all employers to integrate mental health education

**Data availability statement:** All relevant data are within the manuscript and its Supporting Information files.

**Funding:** The author(s) received no specific funding for this work.

**Competing interests:** The authors have declared that no competing interests exist.

into employee ideological and political education to enhance national mental health literacy. However, by the end of 2019, China's overall depression prevalence had reached 2.1%, while anxiety disorders affected 4.98% of the population, with workers' mental health being particularly concerning. In response, the General Office of the State Council issued the"14th Five-Year Plan for National Health" in 2022, emphasizing strengthened mental health services for the working population to promote social stability, harmonious relationships, and public well-being. Regrettably, in the same year, a national occupational health literacy survey conducted by China's National Health Commission—which monitored work-related stress, insomnia, depression, and anxiety among workers—revealed that 15% of surveyed laborers reported experiencing negative emotions such as depression, tension, or anxiety in the past year, ranking third among 16 health issues examined across 23 industries. These findings underscore the severity of mental health challenges among Chinese workers today.

According to experiences from developed countries, public mental health is most vulnerable during economic transitions [1]. Under the "new normal" of economic development, China's slowing economic growth has become an inevitable trend [2], exacerbating labor market volatility and increasing uncertainties for workers, thereby constraining their career advancement. Fang et al. [3] compared the "age-income" trajectories of workers in China and the U.S. over the past 30 years and found that in China, the peak earning age dropped from 55 to around 35, with cohort effects outweighing experience effects—indicating a sharp decline in returns to work experience. The weakening of accumulated experience-based advantages suggests a rapid contraction of the prime period for career advancement. The 2023 China Youth Workplace Survey reveals that over 50% of young workers express anxiety about career promotion, particularly when opportunities are limited, feeling their efforts are inadequately rewarded. More than 65% report frequent work-related anxiety, with 30% experiencing severe workplace stress in the past year. This anxiety stems not only from external competition but also from unmet personal expectations, challenges to self-worth, and peer-comparison pressures. Against this backdrop, a "lying flat" (tang ping) phenomenon has emerged among Chinese youth [4]—where young workers settle for the status quo, abandoning intense competition to alleviate stress. Thus, as career advancement uncertainties grow, critical questions arise: Will workers persist in striving, potentially compromising their mental health, or will the "lying flat" culture lead them to reduce self-imposed stress? This issue warrants further examination. Investigating how career advancement affects workers' mental health can contribute to enhancing China's overall health human capital.

Current research predominantly focuses on the impact of labor supply on workers' health outcomes, with most studies limited to correlational analyses [5, 6]. Few scholars have examined the issue from the perspective of career advancement—specifically, whether career advancement constraints (i.e., heightened future uncertainty) trigger behavioral changes at the micro-level, thereby impairing workers' mental health. This study employs panel data from the 2018 and 2020 China Family Panel Studies (CFPS) to empirically analyze the impact of career advancement on workers' mental health. We clarify this relationship through rigorous causal inference, utilizing

mediation effect analysis to test underlying mechanisms and instrumental variables to address endogeneity concerns. Our findings not only highlight the role of uncertainty in shaping mental health outcomes but also help bridge the gap between macro-level policies and micro-level individual perceptions, ultimately contributing to improved worker well-being.

## Literature review

Existing research has paid scant attention to the direct effects of career advancement on workers' health outcomes. The literature most closely related to this study primarily focuses on two aspects: the influencing factors of career advancement and the determinants of workers' mental health.

Existing research has extensively examined the influencing factors of workers' career advancement and employment quality, with most studies focusing on individual characteristics. From a gender perspective, studies consistently show that women's employment quality and labor market performance are significantly lower than men's [7, 8]. In terms of human capital, workers with higher education levels and greater skills tend to achieve better employment quality and higher labor income [9]. As a crucial human capital investment, health significantly impacts career advancement. Better physical health enables workers to obtain higher labor income and better employment performance [10, 11], while poor mental health substantially reduces labor participation and hinders career advancement [12]. Additionally, age and work experience have traditionally served as important evaluation criteria for career advancement. Earlier studies suggested that accumulated work experience positively influences career advancement as workers age [13]. However, recent findings by Fang et al. [3] reveal a notable contrast: while American workers' "prime career advancement period" remains stable at 45–50 years old, China's has declined from 55 to around 35 years old. This suggests that in today's evolving socioeconomic context, age may increasingly become an obstacle to career advancement. Overall, while existing literature has thoroughly investigated the factors influencing workers' career advancement, it has paid insufficient attention to the consequences of constrained career progression.

Studies on the determinants of workers' health outcomes have primarily focused on both macro-level and micro-level individual factors. At the macro level, economic downturns negatively impact workers' mental health as they develop pessimistic expectations about future employment prospects [14]. Moreover, adequate social support and enhanced social equity help individuals better cope with life difficulties, reducing negative emotions caused by life stress and thereby alleviating psychological distress [15, 16]. At the micro level, rural residents demonstrate higher susceptibility to depression compared to urban dwellers [17], while women exhibit greater vulnerability to depressive moods than men [18]. Young adults facing career and educational pressures typically report lower mental health levels [19]. Regarding educational attainment, highly-educated workers generally possess stronger health awareness, better health literacy, and superior cognitive abilities to self-regulate negative emotions [20]. From a labor perspective, excessive working hours adversely affect both physical and mental health. Prolonged work schedules reduce time available for exercise and sleep, ultimately deteriorating health outcomes and exacerbating depressive symptoms [21, 22]. Workplace stress-induced anxiety and depression progressively accumulate fatigue levels among workers [23–25]. Most existing studies on workers' mental health rely on correlational analyses and utilize samples with excessively broad age ranges, failing to adequately address endogeneity concerns [26, 27].

In summary, this study adopts a novel uncertainty perspective to investigate how workers' career advancement affects their mental health, with particular emphasis on elucidating the underlying mechanisms. By employing instrumental variable approaches to address endogeneity concerns, our research aims to fill critical gaps in the existing literature.

## Theoretical analyses and hypotheses

Career advancement typically promises higher future earnings and enhanced social status. However, the accompanying promotion pressure may induce workers to adopt unhealthy behaviors. According to the theory of extended compensation [28], workers often prolong their working hours to secure greater remuneration and obtain career

advancement opportunities such as promotions or training. Yet workers face finite time resources, creating a trade-off between work and leisure. Prolonged working hours can lead to overexertion, which detrimentally affects physical health [29] and significantly compromises overall physiological well-being [30, 31]. Notably, deteriorating physical health further undermines mental health [32, 33]. Consequently, when confronted with career advancement constraints, workers are more likely to extend working hours in pursuit of higher earnings and development opportunities—a process that may trigger declines in physical health. Based on this reasoning, we propose the following hypothesis:

### Hypothesis 1

Career advancement constraints will lead to prolonged working hours, which in turn deteriorate physical health and ultimately impair mental well-being.

The theory of relative deprivation posits that disparities between individuals can lead workers to experience cognitive gaps characterized by a "others-have-but-I-don't" perception [34]. On one hand, the sense of "inequality" stemming from relative deprivation generates frustration and stress, eroding confidence in the future and consequently impairing mental health [35, 36]. On the other hand, when workers persistently face disadvantaged competitive positions, their upward mobility aspirations and emotional dissatisfaction [37] significantly diminish the efficacy derived from career advancement. This manifests as destructive voice behaviors – disparaging remarks about work and organizations that foster toxic work environments and negative values [38], further exacerbating adverse mental health effects [39].Career advancement inherently involves competition among workers. Therefore, when workers face career advancement constraints, the resulting anxiety and negative perceptions may not only distort positive work attitudes but also undermine future confidence, ultimately compromising mental health [40].

### Hypothesis 2

Career advancement constraints are more likely to induce negative work values and loss of future confidence among workers, thereby adversely affecting their mental health.

## Materials and methods

### Data sources

This study utilizes panel data from the 2018 and 2020 waves of the China Family Panel Studies (CFPS). Initiated in 2010, the CFPS covers tens of thousands of households across 25 provinces and municipalities, with biennial follow-up surveys. The data boasts extensive national representativeness, employing stratified and multi-stage sampling methods that capture approximately 95% of China's total population [41]. Moreover, the dataset comprehensively documents various dimensions of Chinese society, including economic and demographic characteristics, while providing detailed measures of core variables such as health status, health behaviors, and job promotion satisfaction – making it particularly suitable for this investigation. Given our focus on career advancement's impact on workers' mental health, we restrict our analysis to individual-level CFPS samples. To construct our working sample, we exclude students, agricultural workers, the unemployed, and those who have exited the labor force. Recognizing that substantial age variation may introduce heterogeneity in career advancement as stages – potentially causing self-selection bias and endogeneity concerns – we further restrict our sample to workers aged 30–40. Career advancement theory [42] suggests this age cohort is typically in the establishment stage, where securing promotions and achieving upward mobility constitute primary professional objectives. Additionally, this narrower age range ensures respondents face relatively homogeneous work-life challenges during our study period, enhancing estimation accuracy. After eliminating observations with missing values, our final analytical sample comprises 2,316 individuals.

## Variable selection

**Dependent variable.** The dependent variable in this study is workers' self-rated mental health status, measured as a continuous variable. The CFPS employed the 8-item Center for Epidemiologic Studies Depression Scale (CESD-8) in both 2018 and 2020 survey waves to assess individuals' subjective psychological states. Specifically, respondents were asked to rate their frequency of experiencing the following symptoms: (1) feeling depressed, (2) difficulty concentrating, (3) sleep disturbances, (4) feeling happy (reverse-coded), (5) feeling lonely, (6) feeling sad, and (7) perceiving life as overwhelming. Higher composite scores indicate more severe depressive symptoms and consequently poorer mental health status.

**Independent variables.** The independent variable in this study is workers' "career advancement status." Career advancement theory [42] posits that the essence of career advancement lies in the formation of workers' self-concept. Workers' self-concept comprises both perceived self and ideal self [43], where the former refers to self-evaluation based on current circumstances, while the latter represents the desired state workers aspire to achieve. When the perceived self and ideal self cannot be reconciled, this significantly impacts career advancement [44]. Promotion, as a critical component of individual career advancement, influences workers' life planning through associated improvements in compensation and benefits [45]. Consequently, when workers express dissatisfaction with current promotion opportunities or face a lack of advancement prospects, the discrepancy between perceived self and ideal self indicates constrained career advancement. Based on this framework, the study employs "promotion satisfaction" as a proxy variable to measure the disparity between ideal self and actual self, serving as an indicator of career advancement status. Specifically, workers reporting being "very dissatisfied" or "dissatisfied"with promotions or facing "no promotion opportunities" are classified as experiencing career advancement constraints (coded as 1). Conversely, those reporting being "very satisfied" or "satisfied" are considered to have unconstrained career advancement (coded as 0), creating a dummy variable for analysis.

**Control variables.** This study controls for key individual and household characteristics that may influence career advancement. At the individual level, we include: gender, age, income, education level, place of residence, number of children, marital status, and whether the individual holds a civil service position [46, 6]. These variables account for physiological characteristics, human capital endowments, and job characteristics that prior research identifies as significant determinants of career progression. For household characteristics, we incorporate per capita household income and household size to capture resource constraints and dependency burdens that may exacerbate worker stress [47]. The specific coding schemes for all control variables are detailed in Table 1.

**Instrumental variables.** Considering that workers' mental health levels can, in turn, affect their perception of career advancement, this may lead to an endogeneity issue of reverse causality. To address this, we will adopt the concept of "peer effects" to construct instrumental variables [48], that is, calculating the average perception of career advancement among workers with the same "province, age, gender, education level, and income level", as an instrumental variable for individual workers' career advancement. On the one hand, these workers with similar characteristics are more likely to be homogeneous groups; the better the career advancement prospects of a group, the more likely it is that an individual worker, as a member of that group, will also have better career advancement prospects. On the other hand, the career advancement status of similar groups does not directly affect the mental health of individual workers; it only affects their mental health by influencing their possibilities for career advancement, thus making this instrumental variable reasonable.

**Mechanism variables.** The variables related to the mechanism test in this paper are as follows: workers' working hours, deterioration in health status, work values and workers' confidence in the future.

## Model setting

Since the dependent variable in this paper is a continuous variable and uses two periods of CFPS panel data from 2018 and 2020, a fixed-effects model is constructed to perform OLS regression:

$$Depression\ index_{itp} = \alpha + \beta Career\ advancement_{itp} + \eta X_{itp} + \alpha_p + \varepsilon_t + u_{itp} \tag{1}$$

**Table 1. Variable definitions and types.**

| Variable name | Variable definition |
|---|---|
| *Dependent variable* | |
| Depression index | Self-rated scores in the CESD-8 questionnaire: |
| | Continuous variable |
| *Independent variable* | |
| Career advancement | Respondent is not satisfied with promotion or has no opportunity for promotion: Yes＝1 |
| | No＝0 |
| *Control variables* | |
| Gender | Male＝1 |
| | Female＝0 |
| Age | The year of the questionnaire minus the year of birth gives the age: |
| | Continuous variable |
| Income | Monthly total income of the worker; Logarithmic processing |
| Education of worker | Illiterate＝1 (i.e., no education) |
| | Primary school＝2 |
| | Junior high school＝3 |
| | Senior high school＝4 |
| | Associate degree＝5 |
| | Bachelor's degree＝6 |
| | Master's degree＝7 |
| | Doctoral degree＝8 |
| Place of residence | Rural＝1; Urban＝0 |
| Marital status | No spouse (including divorced, widowed, and never married) = 0 |
| | Spouse (including married living with a spouse, married but not temporarily living with a spouse, separated)= 1 |
| Number of children | The number of children currently owned by the worker: Continuous variable |
| Occupational nature | Whether the worker is a civil servant: Yes＝1 |
| | No＝0 |
| Family size | The total number of co-residing family members. |
| Per capita family income | The ratio of total family income to the total number of family members: |
| | Continuous variable |
| *Mechanism variables* | |
| Work hours | The number of hours worked per week by the worker: |
| | Continuous variable |
| Health status change | Improved＝1, Same (no change) = 2, Worsened＝3 |
| Work values | Agreement with "Interpersonal relationships are more important than real talent": Strongly disagree＝1 |
| | Disagree＝2 |
| | Neutral＝3 |
| | Agree＝4 |
| | Strongly agree＝5 |
| Confidence in the future | Very low confidence＝1 |
| | Low confidence＝2 |
| | Neutral＝3 |
| | High confidence＝4 |
| | Very high confidence＝5 |
| Instrumental variable | |
| Career advancement of similar groups(Peer effect) | Calculate the average perception of career advancement among workers with the same province, age, gender, education level, and income level. |

In this context, *Depression index$_{itp}$* represents the depression index of worker *i*, and *Career advancement$_{itp}$* denotes the career advancement status of worker *i*. The coefficient $\beta$ can be interpreted as the extent of the impact of career advancement on the mental health of workers. $X_{itp}$ indicates the control variables, which specifically include variables related to the individual worker and their family characteristics, and $\eta$ represents the regression coefficients of the control variables. $\alpha$ is the constant term. Considering potential economic or cultural differences across various times, regions, $\alpha_p$, $\varepsilon_t$ are set as regional and temporal effects, respectively, and $u_{itp}$ is the random disturbance term.

To avoid the problem of endogeneity, we use a two-stage (2SLS) instrumental variable model:

$$\text{Stage } 1: \quad \textit{Career advancement}_{itp} = \alpha_1 + \beta_1 IV_{itp} + \eta_1 X_{itp} + \alpha_p + \varepsilon_t + \in_{itp} \tag{2}$$

$$\text{Stage } 2: \quad \textit{Depression index}_{itp} = \alpha_2 + \beta_2 \textit{Career advancement}_{it} + \eta_2 X_{it} + \alpha_p + \varepsilon_t + \theta_{itp} \tag{3}$$

In this study, *depression index$_{itp}$* represents the depression index of worker *i*, $IV_{itp}$ denotes the average perception of career advancement among workers with the same "province, age, gender, education level, and income level," which serves as the instrumental variable for individual workers' career advancement. *Career advancement$_{itp}$* represents the career advancement status of worker *i*. The coefficient $\beta_1$ can be interpreted as the extent of the impact of the instrumental variable on workers' career advancement, while $\beta_2$ represents the extent of the impact of career advancement on the depression index after being influenced by the instrumental variable. $X_{itp}$ indicates the control variables, which include specific variables related to the individual worker and their family characteristics, and $\eta_1$ and $\eta_2$ represents the regression coefficients of the control variables. $\alpha_1$ and $\alpha_2$ are the constant terms. Considering potential economic or cultural differences across various times and regions, $\alpha_p$, $\varepsilon_t$ are set as regional and temporal effects, respectively, and $\in_{itp}$ and $\theta_{itp}$ are the random disturbance terms.

## Descriptive statistics

Table 2 provides the statistical description for this study. The statistical description indicates that workers dissatisfied with their career advancement constitute 48.8% of the total sample. By comparing the means of the two groups of variables, it is evident that workers facing career advancement constraints have a higher depression index compared to those with better career advancement, suggesting a lower level of mental health, which is significant at the 1% level. Regarding individual characteristics, workers with career advancement constraints tend to be older, have lower educational levels, have more children, and are more likely to hold rural household registrations, with a lower probability of being public servants. This also reflects the current labor market's inherent age and educational level discrimination, and non-public sector workers report lower satisfaction with their career advancement, potentially due to poorer job stability within this group. There are no significant differences in gender and marital status between the two groups. In terms of family characteristics, workers dissatisfied with their career advancement tend to have larger family sizes and lower per capita income, which may be attributed to greater financial and support pressures within the family, leading to a negative perception of career advancement.

## Results

### Baseline regression results

Table 3 presents the baseline regression outcomes. In column (1), without the inclusion of control variables but controlling for regional and temporal effects, the regression results indicate that for each one-unit increase in the possibility of career advancement constraints, the depression index of workers increases by 0.861, which is significant at the 1% level. In column (2) of the model, control variables related to individual worker characteristics are incorporated; the regression

**Table 2. Descriptive statistics.**

| Variable | Full sample | | Career advancement = 1 | | Career advancement = 0 | | |
|---|---|---|---|---|---|---|---|
| | Mean | Std.Dev. | Mean | Std.Dev. | Mean | Std.Dev. | Difference |
| Depression index | 13.39 | 3.48 | 13.86 | 3.66 | 12.95 | 3.25 | 0.90 *** |
| Gender | 0.57 | 0.50 | 0.58 | 0.49 | 0.57 | 0.50 | 0.02 |
| Age | 34.57 | 3.19 | 34.86 | 3.19 | 34.28 | 3.17 | 0.58 *** |
| Income | 10.65 | 0.94 | 10.47 | 0.96 | 10.82 | 0.88 | −0.34 *** |
| Education of worker | 5.24 | 1.51 | 4.86 | 1.48 | 5.60 | 1.44 | −0.73 *** |
| Place of residence | 0.59 | 0.49 | 0.66 | 0.47 | 0.52 | 0.50 | 0.14 *** |
| Marital status | 0.86 | 0.35 | 0.87 | 0.34 | 0.85 | 0.36 | 0.02 |
| Number of children | 1.42 | 0.75 | 1.47 | 0.77 | 1.37 | 0.74 | 0.10 *** |
| Occupational nature | 0.17 | 0.38 | 0.13 | 0.33 | 0.22 | 0.41 | −0.09 *** |
| Family size | 3.76 | 1.82 | 3.84 | 1.79 | 3.69 | 1.84 | 0.15 ** |
| Per capita family income | 48709.81 | 129978.9 | 38020.77 | 41090.88 | 58894.14 | 176588.42 | −20873.37*** |
| Obs. | 2316 | | 1130 | | 1186 | | |

Notes: The term "Difference" refers to the difference in the mean values between the samples whose occupational career advancement is restricted and those whose career advancement is not restricted. We use t-test to examine whether the differences between the two groups of samples are statistically significant. *p < 0.1. **p < 0.05. *** p < 0.01.

coefficient reveals that for each one-unit increase in the possibility of career advancement constraints, the depression index of workers increases by 0.786, also significant at the 1% level. In column (3) of the model, control variables related to the family characteristics of workers are added again; the regression coefficient shows that for each one-unit increase in the possibility of career advancement constraints, the depression index of workers increases by 0.782, which is significant at the 1% level. After controlling for numerous control variables, although the regression coefficients slightly decrease, they generally maintain a relatively stable trend. By examining the regression results of the control variables, it is evident that workers with a higher number of children tend to have poorer mental health, likely because, in addition to their work commitments, they must also devote a significant amount of time and energy to caring for their children, which may increase their stress. Moreover, workers with lower per capita family income tend to have poorer mental health, possibly because a lower family economic level weakens the workers' ability to cope with uncertainties in life, thereby increasing their depression index.

## Analysis of mechanisms

The baseline results have already indicated that career advancement constraints significantly increase the depression index of workers, thereby exerting a negative impact on their mental health levels. Based on the theoretical analysis in the previous text, this paper will utilize mediation effects to analyze the relevant mechanisms.

**The impact of career advancement on health behaviors and health changes.** This paper uses working hours as a proxy variable for workers' health behaviors; the longer the working hours, the more likely workers are to experience fatigue due to overwork, thereby affecting their mental health levels. This paper uses workers' self-assessed 'change in health status over the past year' as a proxy variable for their physiological health changes. Column (1) of Table 4 below shows the direct impact of career advancement on mental health levels. From column (2), it is evident that career advancement constraints significantly increase workers' working hours; for each one-unit increase in the possibility of career advancement constraints, workers' working hours increase by 4.138, which is significant at the 1% level. From column (3), it is known that restricted career advancement extends working hours; for each one-unit increase in the possibility of career advancement constraints, workers' depression index increases by 0.711, which is significant

Table 3. Results of baseline regression analysis.

| Variables | Depression index | | |
|---|---|---|---|
| | (1) | (2) | (3) |
| Career advancement | 0.861*** | 0.786*** | 0.782*** |
| | (0.145) | (0.148) | (0.148) |
| Gender | | −0.002 | −0.001 |
| | | (0.155) | (0.155) |
| Age | | 0.032 | 0.033 |
| | | (0.024) | (0.024) |
| Income | | 0.019 | 0.024 |
| | | (0.097) | (0.098) |
| Education of worker | | −0.061 | −0.057 |
| | | (0.068) | (0.068) |
| Place of residence | | 0.184 | 0.183 |
| | | (0.171) | (0.171) |
| Marital status | | 0.053 | 0.115 |
| | | (0.111) | (0.118) |
| Number of children | | −1.822*** | −1.732*** |
| | | (0.241) | (0.253) |
| Occupational nature | | −0.153 | −0.149 |
| | | (0.206) | (0.206) |
| Family size | | | −0.071 |
| | | | (0.048) |
| Per capita family income | | | −0.000* |
| | | | (0.000) |
| Constant | 12.9634*** | 13.4545*** | 13.4863*** |
| | (0.122) | (1.252) | (1.260) |
| Year FE | YES | YES | YES |
| Province FE | YES | YES | YES |
| Obs. | 2,381 | 2,381 | 2,381 |
| R-squared | 0.0306 | 0.0617 | 0.0620 |

Notes: In parentheses, standard errors are robust standard errors. *p<0.1. **p<0.05. *** p<0.01.

at the 1% level. From column (4), it is known that for each one-unit increase in the possibility of career advancement constraints, workers' self-assessed health decreases by 0.092, which is significant at the 1% level. From column (5), it is known that career advancement constraints affect workers' depression index by worsening their physiological health; for each one-unit increase in the possibility of career advancement constraints, workers' depression index increases by 0.675, which is significant at the 1% level. In summary, it can be concluded that restricted career advancement may increase workers' depression index by extending their working hours and worsening their health levels, thus validating Hypothesis 1.

**The impact of career advancement on labor values and future confidence.** This paper uses the degree of agreement with the statement 'interpersonal relationships are more important than hard work' as a proxy variable for workers' labor value concepts; the higher the agreement with this view, the more negative the workers' labor values are perceived to be. The paper uses workers' self-assessed 'confidence in the future' as a proxy variable for their evaluation of their own future confidence. Column (1) of Table 5 still represents the direct impact of career

**Table 4. Effect of mechanism variables on work hours and decline in health.**

| Variables | Depression index | Work hours | Depression index | Decline in health | Depression index |
|---|---|---|---|---|---|
| | (1) | (2) | (3) | (4) | (5) |
| Career advancement | 0.782*** | 4.138*** | 0.711*** | −0.092*** | 0.675*** |
| | (0.148) | (0.625) | (0.151) | (0.023) | (0.147) |
| Work hours | | | 0.018*** | | |
| | | | (0.005) | | |
| Decline in health | | | | | −1.159*** |
| | | | | | (0.146) |
| Year FE | YES | YES | YES | YES | YES |
| Province FE | YES | YES | YES | YES | YES |
| Obs. | 2,316 | 2,316 | 2,316 | 2,316 | 2,316 |
| R-squared | 0.079 | 0.228 | 0.084 | 0.026 | 0.108 |

Notes: In parentheses, standard errors clustered at the province level. Standard errors in parentheses. *p<0.1. **p<0.05. *** p<0.01.

**Table 5. Effect of mechanism variables on health level.**

| Variables | Depression index | Work values | Depression index | Confidence in the future | Depression index |
|---|---|---|---|---|---|
| | (1) | (2) | (3) | (4) | (5) |
| Career advancement | 0.782*** | 0.188*** | 0.699*** | −0.173*** | 0.586*** |
| | (0.148) | (0.041) | (0.148) | (0.037) | (0.141) |
| Work values | | | 0.432*** | | |
| | | | (0.073) | | |
| Confidence in the future | | | | | −1.135*** |
| | | | | | (0.093) |
| Year FE | YES | YES | YES | YES | YES |
| Province FE | YES | YES | YES | YES | YES |
| Obs. | 2,316 | 2,316 | 2,316 | 2,316 | 2,316 |
| R-squared | 0.079 | 0.051 | 0.092 | 0.047 | 0.151 |

Notes: In parentheses, standard errors clustered at the province level. Standard errors in parentheses. *p<0.1. **p<0.05 *** p<0.01.

advancement on mental health levels. From column (2), it is evident that career advancement constraints can distort workers' labor values, leading them to believe that maintaining good interpersonal relationships is more important than doing good work, and for each one-unit increase in the possibility of career advancement constraints, the degree of agreement with this view increases by 0.188, which is significant at the 1% level. From column (3), it is known that restricted career advancement can increase the perception of this view; for each one-unit increase in the possibility of career advancement constraints, the depression index of workers increases by 0.699, which is significant at the 1% level. From column (4), it is known that for each one-unit increase in the possibility of career advancement constraints, workers' confidence in the future decreases by 0.173, which is significant at the 1% level. From column (5), it is known that career advancement constraints can affect workers' depression index by reducing their confidence in the future; for each one-unit increase in the possibility of career advancement constraints, workers' depression index increases by 0.586, which is significant at the 1% level. In summary, it can be concluded that restricted career advancement may increase workers' depression index by distorting their labor values and reducing their confidence in the future, thus validating Hypothesis 2.

## Further analysis

**Endogenous test. Instrumental variable.** Considering that workers' mental health levels can also affect their perception of career advancement, that is, when workers have a higher depression index, they may have a more negative evaluation of their own career advancement, which could lead to an endogeneity issue of reverse causality. To address this, we first employ the instrumental variable method to resolve the issue. As mentioned earlier, we calculate the average perception of career advancement among workers with the same "province, age, gender, education level, and income level" known as the "peer effect" as the instrumental variable for this study. Table 6 presents the regression results using the instrumental variable method. From column (1), it can be seen that the career advancement of homogeneous groups is positively correlated with individual workers' career advancement, meaning that the higher the likelihood of career advancement constraints within a homogeneous group, the more likely individual workers' career advancement is to be hindered. From the regression coefficient in column (2) of the model, it can be determined that for each one-unit increase in the likelihood of career advancement constraints, the depression index of workers increases by 0.791, which is significant at the 1% level. This does not significantly differ from the main regression results, thus we can consider our findings to be reliable.

**Propensity score matching inspection.** Considering that the core explanatory variable is constructed from subjective perceptions, and that individuals' perceptions of career advancement may be influenced by other unobservable factors, this could also lead to issues of sample selection bias. To address this, the paper employs propensity score matching (PSM) to resolve the associated endogeneity issues, thereby making the results more robust.

The paper uses PSM to calculate the average treatment effect on the treated (ATT) for workers' career advancement status as follows: Selecting worker characteristics such as gender, age, income, education, household registration, number of children, marital status, public service employment, number of family members, and per capita family income, a logistic regression is performed to estimate the propensity scores, followed by testing the treatment effects through three methods: nearest neighbor matching, kernel matching, and caliper matching. Table 7 shows the matching results, indicating that the average treatment effects from all three matching methods are significant at the 1% level, which is consistent with the main regression results.

Finally, based on the kernel density plots of the treatment and control groups before and after matching, it is evident that the fit after matching is superior to that before matching. This indicates that the method effectively addresses the issue of sample selection bias. See Fig 1 below.

**Table 6. Instrumental variable regression results.**

| Variables | (1) | (2) |
|---|---|---|
| | First stage | Second stage |
| | Career advancement | Depression index |
| Career advancement | | 0.791*** |
| | | (0.150) |
| Peer effect | 1.000*** | |
| | (0.001) | |
| Year FE | YES | YES |
| Province FE | YES | YES |
| Obs. | 2,313 | 2,313 |
| F-value | 58395 | |
| Wald(P-value) | 27.90(0.000) | |

Notes: In parentheses, standard errors are robust standard errors. *p<0.1. **p<0.05. *** p<0.01.

**Table 7. Propensity score matching average processing effect.**

| Matching-method | Treat | Control | ATT | SE | T-value |
|---|---|---|---|---|---|
| Neighbor(n = 1) | | | | | |
| Depression index | 13.848 | 13.910 | 0.938 | 0.194 | 4.16*** |
| Kernel | | | | | |
| Depression index | 13.848 | 13.085 | 0.763 | 0.153 | 4.96*** |
| Caliper(0.01) | | | | | |
| Depression index | 13.848 | 13.113 | 0.734 | 0.156 | 4.69*** |

Notes: Matching is only performed on individuals within the common range of values.*p < 0.1 **p < 0.05 *** p < 0.01.

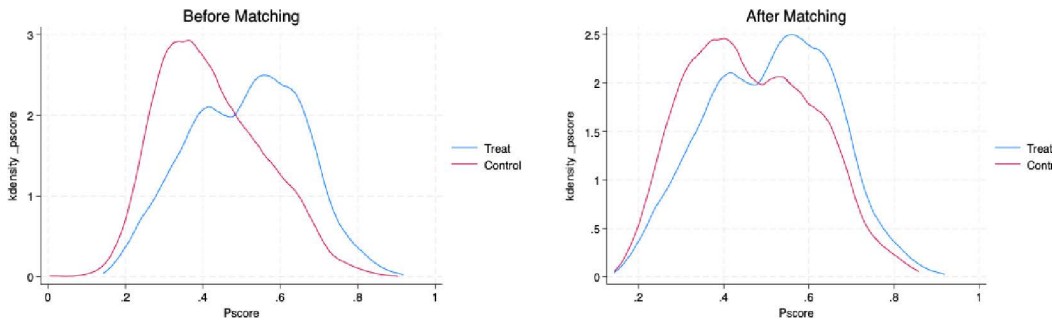

**Fig 1. Kernel density matching plot for PSM.**

**Analysis of heterogeneity. Heterogeneity in gender.** Career advancement paths may differ among workers of different genders, hence the paper conducts separate regressions for male and female workers. Fig 2 shows that for male workers, a one-unit increase in the likelihood of career advancement constraints leads to a rise in the depression index of 1.23, which is significant at the 1% level, but this effect is not significant for female workers.

**Heterogeneity in income level.** Considering that workers with different income levels may perceive career advancement differently, the paper categorizes workers' income into low, medium, and high income groups, and income levels based on the median income of the sample, defining those with worker income below the third percentile, between the third and seventh percentiles, and above the seventh percentile' as low-income, medium-income, and high-income

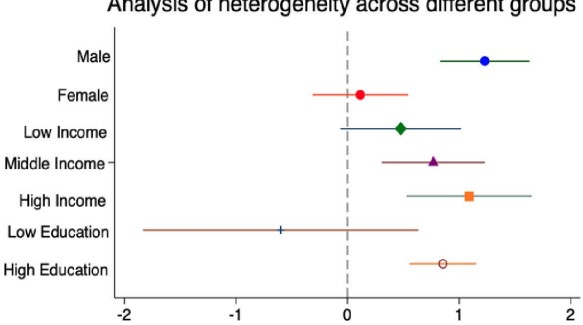

**Fig 2. Heterogeneity analysis by group.**

groups, respectively. The regression results indicate that career advancement constraints mainly increase the depression index of medium and high-income workers. For each one-unit increase in the likelihood of career advancement constraints, their depression index increases by 0.738 and 1.089, respectively, both significant at the 1% level. However, there is no significant impact on the mental health levels of low-income workers.

**Heterogeneity in education level.** Taking into account that workers with different educational levels may have different expectations for career advancement and may start at different career points, the paper conducts separate regressions for workers with different educational levels. Workers who have completed only compulsory education or less are considered to have a low level of education, while those with education beyond the compulsory level are regarded as having a higher level of education. Fig 2 shows that career advancement constraints increase the depression index of workers with higher education levels. For each one-unit increase in the likelihood of career advancement constraints, their depression index rises by 0.855, but this effect is not significant for workers with lower educational levels.

## Conclusions and discussion

This paper utilizes panel data from the China Family Panel Studies (CFPS) for the years 2018 and 2020, employing OLS regression analysis to first examine the impact of workers' career advancement on their mental health levels. Secondly, by logically organizing existing literature and viewpoints, we have formulated relevant hypotheses regarding the underlying mechanisms and tested them using mediation effects. The paper also considers endogeneity issues within the model and employs instrumental variable methods and propensity score matching (PSM) to address these concerns. In addition, the paper conducts heterogeneity analyses for workers of different genders, income levels, and educational levels to explore how career advancement constraints among different groups may affect mental health. Finally, we also perform robustness checks on the research findings by adding control variables and expanding or excluding samples. Against the backdrop of increasingly fierce competition in the labor market, it is hoped that our study will raise societal awareness of the mental health levels of young workers, which has significant practical implications for improving the level of healthy human capital in China.

Firstly, the baseline regression results indicate that restrictions on workers' career advancement lead to a significant increase in their depression index. The essence of career advancement constraints is the uncertainty workers face about the future. Existing literature has shown that uncertainty and the negative beliefs it induces are significant factors in causing anxiety and depression [49], which is consistent with our research findings. As the unemployment rate among young Chinese workers gradually increases and the uncertainty in the labor market grows, workers may find it difficult to achieve more satisfactory career advancement. Since early adulthood is the prime period for career advancement, constraints in career advancement imply a narrowing of upward mobility opportunities, which may increase the depression and anxiety of young workers, thereby reducing their mental health.

Secondly, the main mechanisms of the paper are twofold. On one hand, career advancement constraints can lead to extended working hours for workers and a deterioration in their health status, thereby increasing their depression index and reducing their mental health levels. Current research also indicates that extended working hours can cause symptoms of depression and anxiety among workers [50], which aligns with our study's conclusions. On the other hand, the inequality caused by career advancement constraints can enhance workers' sense of relative deprivation, leading to discontented subjective emotions [51, 52]. This distorts workers' labor values, leading to more negative cognitions. Moreover, career advancement constraints imply a narrowing of upward mobility opportunities, which to some extent undermines workers' confidence in the future. Insufficient confidence can also lead to an increase in their depression index [53–55]. It is evident that longer working hours, deteriorated health conditions, and negative value concepts can also have a certain impact on workers' mental health.

Lastly, this paper conducts heterogeneity analysis based on individual characteristics of workers (gender, income, and education level). The results reveal that compared to women, career advancement constraints primarily increase

the depression index of men, meaning that it more significantly affects the mental health of male workers. This may be due to the influence of male breadwinner culture, in which men in East Asian cultures are expected to bear more family financial responsibilities. For men, the stigma associated with setbacks in career advancement is stronger than for women [56], and career advancement constraints imply difficulty in improving future income levels, which may increase their psychological stress [57]. Current findings on the impact of income levels on depressive symptoms are relatively consistent, indicating that those with lower incomes have lower mental health levels [58, 59], while the impact of education level on workers' mental health is inconclusive. On one hand, some studies suggest that a higher level of education can promote individual mental health [60], and on the other hand, some scholars find through correlation analysis that there is no significant relationship between the two [61]. Our study finds that compared to low-income and low-educated workers, career advancement constraints mainly elevate the depression index of workers with medium and high income levels and higher education levels. This may be because low-income, low-educated workers are often in more grass-roots positions or are more likely to be in a state of career instability, whereas middle and high-income groups often have better development conditions at work, making it easier for them to meet the conditions or thresholds for career advancement. The abundance of career advancement opportunities makes them more sensitive to the state of their career advancement.

Based on the above findings, the paper proposes the following recommendations: Firstly, the government should strengthen the regulation of the labor market to prevent excessive labor and overtime, and further urge enterprises to provide appropriate medical insurance and psychological counseling services for workers to promote their health protection. Secondly, the government should enhance guidance on public opinion and urge social media to promote positive concepts of labor and human capital, which will help to mitigate social biases, especially in the labor market, against 'age discrimination' and help young workers regain confidence in self-improvement. Lastly, the government should give more policy preference to workers with low income and education levels, for example, by providing them with skills training, granting subsidies for skill learning, and enhancing their ability to improve labor skills, which will help them build self-confidence and is also beneficial for improving the health human capital levels of vulnerable groups in the labor market.

## Limitations

This study also has certain limitations. Firstly, although this paper uses panel data from two periods of CFPS in 2018 and 2020, the short time span makes it difficult to estimate the long-term impact of career advancement on workers' mental health. Secondly, to alleviate endogeneity issues, the paper has narrowed the age range of the sample as much as possible, but this has also resulted in the disadvantage of a smaller sample size. Lastly, since both the independent and dependent variables in this paper are subjective, although the author has employed instrumental variable methods and propensity score matching to mitigate reverse causality and sample selection issues, there may still be some omitted variables, which means that the endogeneity problem cannot be completely eliminated. In future research, we will incorporate more updated data and construct more precise variable indicators to enhance the reliability of the research findings.

## Supporting information

**S1 File.**
(XLSX)

## Author contributions

**Conceptualization:** Hanwei Li, Xiaoheng Wu.

**Funding acquisition:** Hanwei Li.

**Writing – original draft:** Xiaoheng Wu.

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
