## [Decision Letter · Decision Letter 0]

Dear Dr. Xiaoheng,

Thank you for submitting your manuscript to PLOS ONE. After careful consideration, we feel that it has merit but does not fully meet PLOS ONE’s publication criteria as it currently stands. Therefore, we invite you to submit a revised version of the manuscript that addresses the points raised during the review process.

We look forward to receiving your revised manuscript.

Kind regards,

I Gede Juanamasta

Academic Editor

PLOS ONE

 [Wu and Li thank National Social Science Foundation of China: Research on Building a Healthy China, No. 16BJY001]. 

Reviewers' comments:

Reviewer's Responses to Questions

**Comments to the Author**

1. Is the manuscript technically sound, and do the data support the conclusions?

Reviewer #1: Partly

Reviewer #2: Partly

2. Has the statistical analysis been performed appropriately and rigorously?

Reviewer #1: Yes

Reviewer #2: N/A

3. Have the authors made all data underlying the findings in their manuscript fully available?

Reviewer #1: No

Reviewer #2: No

4. Is the manuscript presented in an intelligible fashion and written in standard English?

Reviewer #1: No

Reviewer #2: Yes

Reviewer #1: Abstract:

o Language is somewhat repetitive (e.g., repeated phrases like “development restriction”).

o The final sentence regarding mental health differences between low and high income groups is confusing and repetitive.

o The abstract should explicitly state why this research is important, briefly mention implications or recommendations.

Introduction:

o Redundant phrasing ("three-piece work set" repeated unnecessarily).

o The paragraph structure is somewhat unclear, blending government policy context and empirical health statistics without sufficient transition.

o Missing explicit statement of the research gap that this paper will address in the Introduction itself.

Literature Review:

o Several awkwardly phrased sentences (e.g., overly complex and long sentences).

o Insufficient synthesis: the literature review currently feels like a series of unrelated studies rather than a coherent narrative.

o Little international comparison or broader theoretical context provided.

Theoretical Analysis and Hypotheses:

o Theories are presented clearly, but could benefit from further critical reflection or acknowledgment of limitations.

o Some sentences are too long or convoluted, making them hard to follow.

Models and Variables:

o Model description contains minor language/grammatical issues (e.g., confusing notation).

o Justification for sample selection (ages 30-40) is clear but could be briefly re-emphasized.

o Explanation of why certain control variables are included could be more explicit.

Statistical Description:

o Minor grammatical issues (e.g., "people with limited development accounted for…").

o Insufficient explanation of why certain descriptive statistics are important or meaningful.

Empirical Analysis and Results:

o Limited explanation of unexpected findings (e.g., negative relationship between high education and health).

o R² values are relatively low, and the manuscript does not discuss the implications of this sufficiently.

Endogeneity Issues:

o Explanation of PSM approach is excessively detailed in footnotes, disrupting readability.

o Presentation of kernel density plots could be improved visually.

o The reasoning behind choosing "35-year-old threshold" is clear but could benefit from additional theoretical framing.

Mechanism Testing and Heterogeneity Analysis:

o Some results lack sufficient theoretical explanation (e.g., why working hours significantly impact mental health but not physical health).

o Slight repetition in explaining expected results and interpretations.

Robustness Tests:

o Limited explanation about why specific robustness checks were selected.

o Tables lack detailed interpretation, making them seem more routine than analytical.

Conclusions:

o Recommendations are somewhat generic and could be more specific (e.g., suggesting how to practically enhance labor supervision).

o Limited explicit reflection on study limitations or directions for future research.

References:

o Minor inconsistencies in formatting (e.g., missing years for some references).

o Some cited sources in the text appear to be missing or inconsistently listed in the bibliography.

Overall Recommendations:

1. Thorough proofreading and language editing for improved clarity, readability, and consistency.

2. Better integration and synthesis of literature to clearly highlight research gaps and theoretical contributions.

3. Clearly link empirical findings back to theoretical frameworks throughout the manuscript.

4. Provide deeper interpretive insights, explain unexpected results explicitly, and clearly articulate practical implications and limitations.

5. Please kindly check journal guidelines to follow manuscript structure.

Reviewer #2: The topic of the manuscript is interesting, but several key methodological and other weaknesses need to be addressed before the manuscript can be published.

Introduction

The state of the literature is not adequately described, and there are hardly any references to international literature. The section on gender (From a gender perspective....) on page 2 lacks references to the literature, for example.

Hypotheses

Two hypotheses are put forward, but these are not the focus of the discussion. Instead, new questions and analyses are repeatedly presented, e.g. on heterogeneity analyses, differentiations between inside/outside the system, gender and income. These analyses come as a surprise and were not introduced earlier.

Methods

1. A methods section is missing, in which the data, the variables used and all methods employed are described. Some methods are described in the results section, some in footnotes. Please describe the data, all methods (including benchmark regression and matching methods) and all variables in the methods section. For example, it is not yet clear what is meant by ‘inside/outside the system’ or ‘whether there is a staff’.

2. The methods used do not allow the hypotheses to be tested. The hypotheses are questions that need to be answered using mediation analysis. For example, the question is whether limited career opportunities have a negative impact on subjective health via the negative consequences of working hours and health behaviour. In this sense, health behaviour, for example, represents a mediator of the relationship between career opportunities and health. However, mediation analyses are not calculated. Nevertheless, mechanisms are discussed without being investigated. Hypothesis 2 with relative deprivation is not answered.

3. Only cross-sectional data is available, which does not allow any conclusions to be drawn about the direction of the relationship. It may also be the case that poor health influences career development. This problem is not addressed; instead, the influence of career opportunities on health is always discussed. Correctly, no statements can be made about the direction of the effect; only the relationship between the two variables can be discussed.

Results

1. The tables have inadequate titles and labels for the rows and columns. Information is missing, e.g. about what is meant by ‘highly educated’. Please clarify this.

2. Many sensitivity analyses are performed due to the endogeneity problem. Although it is appropriate to address this problem, it makes it difficult to identify the common thread running through the results. It is suggested that these sensitivity analyses be moved to the appendix and that the results section focus on answering the hypotheses. Therefore, it is also recommended that the robustness test be moved to the appendix.

3. It is rather unusual to provide explanations for the findings in the results section. This is usually done in the subsequent discussion.

4. The question arises as to why career opportunities are referred to as a treatment effect. What is the treatment effect in this case? Different terms are also used for this variable career opportunities, such as promotion block, non-promotion, limited development or constrained workers. This is confusing; please use consistent terminology.

5. There may be an error in Table 2. Is it really the case that non-promotions have better health (higher values)?

6. As mentioned above, no mechanisms are tested, only bivariate correlations that are linked theoretically but not empirically. Mediation analysis, in particular causal mediation analysis, represents an appropriate empirical method for testing the mechanisms. This method should be used.

7. The discussion fails to place the findings in the context of previous research. No other studies that have come to similar or different conclusions are cited in the discussion. This should be done.

**Do you want your identity to be public for this peer review?** For information about this choice, including consent withdrawal, please see our Privacy Policy

Reviewer #1: **Yes: ** Yupin Aungsuroch

Reviewer #2: No

---

## [Author Response · Author response to Decision Letter 1]

20 May 2025

Dear Reviewers and Editors,

Thank you for your comments. The authors have made revisions according to each of your suggestions. Given the substantial number of changes, we have compiled all the revision details in a document titled "Response to Reviewers." We kindly ask for your further review and comments.

Thank you once again!

---

## [Decision Letter · Decision Letter 1]

Dear Dr. Wu,

Thank you for submitting your manuscript to PLOS ONE. After careful consideration, we feel that it has merit but does not fully meet PLOS ONE’s publication criteria as it currently stands. Therefore, we invite you to submit a revised version of the manuscript that addresses the points raised during the review process.

We look forward to receiving your revised manuscript.

Kind regards,

I Gede Juanamasta

Academic Editor

PLOS ONE

Reviewers' comments:

Reviewer's Responses to Questions

**Comments to the Author**

Reviewer #1: All comments have been addressed

Reviewer #3: All comments have been addressed

2. Is the manuscript technically sound, and do the data support the conclusions?

Reviewer #1: Yes

Reviewer #3: Yes

3. Has the statistical analysis been performed appropriately and rigorously?

Reviewer #1: Yes

Reviewer #3: N/A

4. Have the authors made all data underlying the findings in their manuscript fully available?

Reviewer #1: No

Reviewer #3: Yes

5. Is the manuscript presented in an intelligible fashion and written in standard English?

Reviewer #1: Yes

Reviewer #3: Yes

Reviewer #1: The authors addressed all the comments. I have no more comments.

The manuscript is well-prepared. It is ready to be published.

Reviewer #3: to me, this paper have simple conclusion which show promotion restriction will lead to bad mental health. this is logical.

however, the complex statistics with lack of diagram have lead me to puzzle.

i recommend reduce statistics with better diagrammatical presentation. otherwise, this is a complex paper with minimal interest to readers for my understanding.

**Do you want your identity to be public for this peer review?** For information about this choice, including consent withdrawal, please see our Privacy Policy

Reviewer #1: **Yes: ** Yupin Aungsuroch

Reviewer #3: **Yes: ** tan jih huei

---

## [Author Response · Author response to Decision Letter 2]

21 Jun 2025

We are extremely grateful for the valuable comments provided by the reviewer. The authors have taken your concerns very seriously and have discussed this issue thoroughly with our collaborators at the earliest opportunity. As you pointed out, we have employed a substantial amount of statistical analysis to demonstrate the rationality and accuracy of our research, which may indeed be complex for some readers. In response to your suggestion to “reduce statistics with better diagrammatical presentation,” we have added illustrations to the PSM section to more intuitively show the differences before and after sample matching. Additionally, we have replaced the regression tables in the heterogeneity analysis section with figures, allowing for a more intuitive comparison of coefficient differences across different groups. We have also removed the robustness test section to mitigate the negative reading experience caused by excessive statistical analysis.

In addition to making the aforementioned changes, please allow the authors to explain the reasons for retaining most of the current statistical content:

1. Baseline Regression: This is the fundamental part of our investigation into the impact of career advancement on mental health. It directly demonstrates whether the conclusion of our research question holds true. Some of the control variables and fixed effects are included to meet the standard requirements of demographic econometric models, and thus, they need to be retained.

2. Mechanism Analysis: This section aims to explain why career advancement can affect workers' mental health, that is, to inform readers through which factors career advancement influences mental health. We have employed mediation effect analysis here, which is not only the most common and classic method in mechanism analysis but also a frequently used approach for mechanism testing in the PLOS ONE journal[1][2]. We will attach these references for you to verify our statements. Moreover, another reviewer explicitly requested the use of mediation effect methods, so we need to retain this section.

3. Endogeneity Issues: This is a crucial issue in our research field. We have successively employed instrumental variable and PSM methods. The instrumental variable is used to address the reverse causality problem. Specifically, both our dependent and independent variables are subjective. Career advancement constraints may lead to a decline in mental health. Conversely, poor mental health can also cause workers to perceive their career advancement negatively. If this reciprocal relationship is not addressed, it becomes difficult to determine whether the independent variable affects the dependent variable or vice versa, leading to biased estimation results. The instrumental variable is the best way to resolve this issue, and thus, it needs to be retained. PSM is used to address the problem of sample selection bias. It constructs a counterfactual framework to make the groups in the treatment and control groups more comparable, thereby more clearly identifying the impact of career advancement on workers' mental health and ensuring the accuracy of the estimation results. Regarding the complex statistical methods you mentioned, which are mainly in this section, we have considered whether to remove them. However, after referring to highly relevant articles published in PLOS ONE within the same research field as our study, we found that both methods are common and necessary. Therefore, we kindly ask the reviewer for understanding.

4. Robustness Test: This section tests the reliability of our conclusions through a series of criteria by changing the sample selection and model construction methods. Since this is not the core content of our paper, we are willing to remove it.

5. Heterogeneity Analysis: This section aims to explain why different groups show inconsistent changes in mental health after being affected by career advancement. It enriches our research and is a part that is almost always included in studies within this field. Considering the previous excessive statistical tables, we have replaced them with figures here to allow you to intuitively observe the changes in regression coefficients across different groups.

In summary, we have to retained these statistical analysis sections. We kindly ask for your understanding! Thank you for your understanding.

References

1. Wang C, Guo J, Xu W, Qin S. The impact of digital transformation on corporate green governance under carbon peaking and neutrality goals: Evidence from China. PLoS ONE (v1;2006). 2024;19: 21. doi:10.1371/journal.pone.0302432

2. Zhao X, Shen L, Jiang Z. The impact of the digital economy on creative industries development: Empirical evidence based on the China. Rosak-Szyrocka J, editor. PLoS ONE. 2024;19: e0299232. doi:10.1371/journal.pone.0299232

---

## [Decision Letter · Decision Letter 2]

Research on the impact of career advancement on the mental health of young workers

PONE-D-25-09440R2

Dear Dr. Wu,

We’re pleased to inform you that your manuscript has been judged scientifically suitable for publication and will be formally accepted for publication once it meets all outstanding technical requirements.

Kind regards,

I Gede Juanamasta

Academic Editor

PLOS ONE

Additional Editor Comments (optional):

Reviewers' comments:

Reviewer's Responses to Questions

**Comments to the Author**

Reviewer #1: All comments have been addressed

2. Is the manuscript technically sound, and do the data support the conclusions?

Reviewer #1: Yes

3. Has the statistical analysis been performed appropriately and rigorously?

Reviewer #1: Yes

4. Have the authors made all data underlying the findings in their manuscript fully available?

Reviewer #1: No

5. Is the manuscript presented in an intelligible fashion and written in standard English?

Reviewer #1: Yes

Reviewer #1: Last time I have accepted this manuscript.

I have no more comments. The article is ready to be published.

**Do you want your identity to be public for this peer review?** For information about this choice, including consent withdrawal, please see our Privacy Policy

Reviewer #1: **Yes: ** Yupin Aungsuroch
